# Burnout, Social Comparison Orientation and the Responses to Social Comparison among Teachers in The Netherlands

**DOI:** 10.3390/ijerph192013139

**Published:** 2022-10-12

**Authors:** Abraham Pieter Buunk, Veerle Brenninkmeijer

**Affiliations:** 1Department of Psychology, University of Groningen, 9712 TS Groningen, The Netherlands; 2Department of Psychology, Utrecht University, 3508 TC Utrecht, The Netherlands

**Keywords:** burnout, social comparison, identification, affective responses, secondary education

## Abstract

(1) Background: Teaching is a profession in which burnout constitutes a prevalent issue and provides ample opportunity to compare oneself with one’s colleagues, i.e., social comparison. The purpose of this research in the Netherlands was to examine responses to social comparison, in relation to burnout, and to individual differences in social comparison orientation (SCO). (2) Methods: Study 1 employed a retrospective measure to assess responses to social comparison. In Study 2, teachers were confronted with a scenario describing either a well or a poorly functioning colleague. (3) Results: Burnout was associated with more negative responses to well and to poorly functioning colleagues, with a less positive response to well-functioning colleagues. This last effect was fully due to the degree to which one identified or contrasted oneself with this colleague. Especially among high-SCO individuals, identification with a well-functioning colleague was accompanied by a positive affect. High-burnout individuals reported more identification with poorly functioning colleagues, and more contrast and less identification with well-functioning colleagues. Responses to well-functioning colleagues were more strongly related to burnout among those high in SCO. (4) Conclusions: An especially negative affect after comparison with well-functioning colleagues is typical for individuals high in burnout, particularly among individuals with a dispositional tendency to compare themselves with others.

## 1. General Introduction

The current prevalence of burnout in our society seems quite high and, for example, in Europe, on average, 10% of the EU workforce reports feeling burned out [1]. The most influential conceptualization of burnout was proposed by Maslach [2] as a syndrome consisting of three aspects (cf. [3]), i.e., emotional exhaustion, the experience of a depletion of emotional resources and feeling ‘empty’ or ‘worn out’; second, depersonalization, a negative, cynical attitude toward one’s work or the recipients of one’s care (e.g., pupils or clients); and third, reduced personal accomplishment, the tendency to evaluate one’s accomplishments at work negatively. Occupational burnout conceptualized as such has been explored in over 13,000 studies and has been found to be associated with, among other things, a lack of external control, neuroticism, workload, role conflict, and a lack of feedback from superiors. Burnout may result in job dissatisfaction, reduced commitment, absenteeism, and turnover (for reviews, see [3,4,5]), and it is associated with health problems such as musculoskeletal complaints [6], and in general with impaired emotional, physical, and social functioning, and more health care use [7]. There also seem to be genetic and epigenetic correlates of burnout, including the DNA methylation of specific regions of the brain-derived neurotrophic factor [8].

One particular sector in which burnout constitutes a serious problem is education. In Europe, it is estimated that 60 to 70% of teachers experience frequent stress and that roughly 30% of teachers show signs of burnout [9]. Problems associated with teacher burnout include time pressure, poor relationships with colleagues, large classes, lack of resources, isolation and lack of support, fear of violence, role ambiguity, poor opportunities for promotion, lack of participation, and behavioral problems of pupils [10,11,12,13].

Teaching is a profession in which there is ample opportunity for social comparison. In the staff room, for instance, teachers exchange information about their lessons and students, thereby revealing information about their functioning. Therefore, in the present research, social comparison processes among individuals with varying degrees of burnout were investigated, in relation to individual differences in social comparison orientation ([13,14], for a review see [15]).

### 1.1. Social Comparison under Stress

In the past four decades, researchers have become increasingly interested in the role of social comparisons with regard to well-being [16], including, in recent years, the role of on-line comparisons [17]. Individuals may engage in *upward comparisons*, comparing themselves with individuals who perform in a more competent and adequate way than they do. Individuals may also engage in *downward comparisons*, comparing themselves with individuals who perform in a less competent and adequate way than they do [18]. According to the Identification–Contrast Model [19], the effects of social comparison depend on whether individuals *identify* or *contrast* themselves with the comparison targets. In the case of identification, individuals tend to focus on the actual or potential similarity between themselves and the comparison target and may regard the other’s position as similar or attainable for themselves. In the case of contrast, individuals use the other as a standard to evaluate their situation. In general, upward identification with a successful colleague may enhance one’s self-image and evoke positive feelings such as hope and admiration, whereas downward identification with an unsuccessful colleague may lower one’s self-image and may produce feelings of worry and fear (e.g., [20,21]). Conversely, upward contrast with a successful colleague may lower one’s self-image and evoke negative feelings such as frustration and resentment, whereas downward contrast may enhance one’s self-image and may produce—at least temporarily—feelings of relief and pride.

It is assumed here that teachers in a state of burnout will be especially sensitive to social comparison, as social comparison processes may overall play an important role among people facing a threat to their well-being (e.g., [22]), and in the context of the working environment, social comparison is a quite prevalent phenomenon (e.g., [23,24,25]). As individuals high in burnout will feel relatively incompetent, they may have lost the ability to identify with, and to be stimulated by, others who are doing better. Hence, individuals high in burnout will show less upward identification and more upward contrast [26], which may over time not help in reducing burnout [27,28].

In sum, based on the reasoning and empirical studies described above, we formulated the following hypothesis:

**Hypothesis** **1.***Burnout is associated with less favorable responses (in terms of affect and identification–contrast) to social comparison*.

### 1.2. Social Comparison Orientation

The last issue examined in the present research concerns the role of individual differences in social comparison tendencies among those experiencing burnout. Making social comparisons may often be a function of one’s personality [29]. Gibbons and Buunk [14] proposed the concept of *social comparison orientation* (SCO) to refer to the personality disposition of individuals who are inclined to base the evaluation of their own characteristics upon comparison with others, who tend to focus on how they are doing in comparison with others, and who have a tendency to relate events that happen to others to themselves (for a recent review, see [15]). Many studies have shown that SCO may act as a predictor and moderator of the effects of social comparisons. In a longitudinal study among nurses, it was found [27] that those who responded with a more negative affect to upward comparisons at T1 showed an increase in burnout at T2 only when they were high in SCO. In the present research, it is predicted that teachers high in social comparison orientation will engage more in all types of social comparisons with others, including identification and contrast. Moreover, the effects of burnout upon identification and contrast and affect are expected to be stronger as teachers are higher in SCO. In sum, we propose the following hypotheses with respect to role of SCO:

**Hypothesis** **2.***SCO is associated with stronger responses (in terms of affect and identification–contrast) to social comparison*.

**Hypothesis** **3.***SCO enhances the associations between burnout and responses to social comparison*.

Responses to social comparison were examined in two studies. Study 1 investigated identification and contrast in response to upward and downward comparison, including the affect involved in these processes, in a retrospective manner. Study 2 was a follow-up experiment in which other teachers were confronted with a scenario about a well-functioning or a poorly functioning teacher, after which, identification, contrast and affect were assessed separately.

## 2. Materials and Methods: Study 1

### 2.1. Participants and Procedure

In Study 1, the sample consisted of 156 teachers in secondary education. The mean age of the participants was 45 years (*SD* = 8.7), and 58% of the sample was male. The average experience as a teacher was 19 years (*SD* = 9.8). The study was conducted in accordance with the Declaration of Helsinki, and extensive formal written informed consent was waived. Participants were approached at their work and were asked to participate in a questionnaire study. When they consented, they were given a questionnaire that they could fill out at home and subsequently return in a postage-free envelope. Anonymity was guaranteed, and participants were informed that they were free to return the questionnaire or not. The procedure in the study was approved by the Netherlands Foundation for Scientific Research (NWO) as part of the research program ‘Fatigue at work’.

### 2.2. Measures

The questionnaire encompassed, among other measures, scales for assessing burnout, social comparison orientation, and identification–contrast.

**Burnout** [13]. Burnout was measured with a Dutch version of the Maslach Burnout Inventory for teachers (MBI-NL-Le, [30]), which consists of three subscales: Emotional Exhaustion, Depersonalization, and Personal Accomplishment (cf., [31]). The internal consistency of the subscales for exhaustion, depersonalization, and personal accomplishment was 0.88, 0.66, and 0.81, respectively. The relatively low internal consistency of depersonalization is in line with several other studies (e.g., [32]). Because the primary interest here was the effects of burnout in general, the personal accomplishment items were recoded, and all the items of the MBI were summed to one burnout measure. Cronbach’s alpha for this total scale was 0.87.

**Social Comparison Orientation** [13]. Social comparison orientation (SCO), the dispositional tendency to compare oneself with others, was measured with the Iowa–Netherlands Comparison Orientation Measure (INCOM; [14,15]). This scale consists of 11 items, such as “I always like to know what others in a similar situation would do” and “I never consider my situation in life relative to that of other people”. The items are responded to on a 5-point scale, ranging from 1 (*strongly disagree*) to 5 (*strongly agree*). The psychometric qualities of the scale are good. Previous research has provided evidence of the construct, discriminant, and concurrent validity of the scale. Its internal consistency is high (Cronbach’s α > 0.80), and its test–retest reliability (over eight months) is 0.72. In the present sample, Cronbach’s alpha was 0.82.

**Identification–Contrast Scale**. This scale was based on a similar scale used in health contexts [33] and consists of four subscales, each containing three items to assess identification and contrast in response to upward and downward comparison.

Items measuring *upward identification* were [13]: 

“When I see teachers who are functioning better than I am, I realize that it is possible to improve”; “When I meet teachers who are performing better than I am, it makes me happy because I realize that it is possible for me to improve”; “When I think of teachers who are functioning better than I am, I have hope that my performance will improve”. 

Items measuring *upward contrast* were: 

“When I see teachers who are doing better than I am, I find it threatening to notice that I am doing not so well”; “When I think of teachers who are functioning better than I am, I feel frustrated about my own performance”; “When I meet teachers who are doing better than I am, I sometimes feel depressed because I realize that I am not doing so well”. 

Items measuring downward identification were: 

“When I see teachers who are doing worse than I am, I experience fear that my performance will decline”; “When I meet teachers who are performing worse than I am, I am sometimes afraid that my future will be similar”;“When I think of teachers who are doing worse than I am, I am sometimes afraid that I will go the same way”. 

Items measuring *downward contrast* were: 

“When I see teachers who are functioning worse than I am, I am happy that I am doing so well myself”; “When I think of teachers who are functioning worse than I am, I feel relieved about my own performance”;“When I meet teachers who are performing worse than I am, I realize how well I am doing”. 

Items could be answered on a 5-point scale ranging from *not at all* (1) to *strongly* (5). The scores on each subscale were computed by taking the mean of the items in the subscale. The internal consistency of the scales was good: Cronbach’s alphas were 0.93, 0.86, 0.85, and 0.83, respectively. Upward identification and contrast did not correlate (*r* = 0.01, *ns*), whereas for downward comparison, the correlation between identification and contrast appeared to be significant (*r* = 0.34, *p* < 0.01).

### 2.3. Statistical Analyses

The data were analyzed with multiple regression analyses. The four subscales in the Identification–Contrast Scale were hierarchically regressed on burnout, social comparison orientation (SCO), and the interaction between these two variables. To reduce the collinearity between the variables and their products, all variables were standardized [13]. Moreover, standardizing variables facilitates the interpretation of results: with standardized variables, the unstandardized regression coefficients reflect the relative contribution of the predictors, controlling for differences in variance [32].

## 3. Results: Study 1

Table 1 and Table 2 show that burnout had significant main effects on upward identification, upward contrast, downward identification, but not on downward contrast. With increasing levels of burnout, individuals identified themselves less and contrasted themselves more with upward comparison targets, and they identified themselves more with downward comparison targets. Hence, Hypothesis 1 was supported with respect to identification–contrast. Furthermore, in line with Hypothesis 2, overall, high-SCO individuals reported more identification and contrast to social comparison.

In addition, burnout and SCO had significant interaction effects on identification and contrast in response to upward, but not in response to downward comparison. The inspection of the simple slopes (see [34]) revealed that, with increasing levels of burnout, there was less upward identification among high-SCO individuals (B = −0.33, *p* < 0.01), but not among low-SCO individuals (B = 0.02, *ns*) (see Figure 1). With increasing levels of burnout, there was also more upward contrast among high-SCO individuals (B = 0.51, *p* < 0.01), and less so among low-SCO individuals (B = 0.25, *p* < 0.01) (see Figure 2). Hence, the prediction that SCO would enhance the association between burnout and the responses to upward and downward comparison (Hypothesis 3) was supported only with respect to upward comparison.

## 4. Introduction: Study 2

Study 1 indicated that teachers high in burnout reported less identification and more contrast in response to upward comparison, especially when they were high in SCO, and overall, reported more identification in response to downward comparison. Because Study 1 was a retrospective study, in addition, an experiment was designed to assess the affective responses to social comparison separately from identification and contrast. That is, in Study 2, participants were confronted with a scenario about a well-functioning or a poorly functioning colleague, after which, positive and negative affect were assessed, as well as the identification and contrast with the comparison target. In this way, the association of identification and contrast with the affective responses to social comparison could be examined.

It must be noted that the experience of negative affect is not necessarily the opposite of the experience of positive affect ([13,35,36], but see also [37]). Positive affect refers to the approach of situations that give an opportunity for reward, and it is associated with obtaining the resources important for survival, such as food and sex. Negative affect implies vigilance, withdrawal, and the avoidance of harmful situations (e.g., [13,38]). As upward identification is aimed at success and improvement, upward identification will result mainly in positive affect. In contrast, downward identification implies a state that should be avoided and is not something one should aim for, which may primarily evoke a negative affect.

In addition to the hypotheses that were addressed in Study 1, it was expected that the more negative affective responses to downward comparison and the less positive affective responses to upward comparison among those high in burnout would be mediated by more identification or less contrast with worse-off others, and by less identification or more contrast with better-off others. Hence, we proposed the following additional hypothesis:

**Hypothesis** **4.***The associations between burnout and the affective responses to social comparison are mediated by identification and contrast*.

## 5. Materials and Methods: Study 2

### 5.1. Participants and Procedure

The sample included 190 teachers in secondary education. No teachers also participated in Study 1. The mean age of the participants was 44 years (*SD* = 9.2), and 60% of the sample was male. The average experience as a teacher was 18 years (*SD* = 9.6). The study was conducted in accordance with the Declaration of Helsinki, and extensive formal written informed consent was waived. Participants were approached at their work and were asked to participate in a questionnaire study. When they consented, they were given a questionnaire that they could fill out at home and subsequently return in a postage-free envelope. Anonymity was guaranteed, and participants were informed that they were free to return the questionnaire or not. The procedure in the study was approved by the Netherlands Foundation for Scientific Research (NWO) as part of the research program ‘Fatigue at work’ (1996–2004).

### 5.2. Measures

**Burnout** [13]. As in Study 1, burnout was measured with the Dutch version of the Maslach Burnout Inventory for teachers (MBI-NL-Le, [30]). The internal consistency of the subscales for exhaustion, depersonalization, and personal accomplishment in this sample was 0.91, 0.61, and 0.84, respectively. Again, the personal accomplishment items were reversed, and then, all items were added up to one burnout measure (Cronbach’s α 0.85). In this sample, the ‘Exhaustion + 1′ criterion (see Study 1) yielded a burnout percentage of 15.3%. 

**Social comparison orientation** [13]. The measure of social comparison orientation (SCO) was the same as in Study 1, i.e., the Iowa–Netherlands Comparison Orientation Measure (INCOM; [14]). In the current sample, Cronbach’s alpha was 0.85.

**Experimental manipulation** [13]. In the second part of the questionnaire, the experimental manipulation was presented. Participants received social comparison information, and their responses to this information were assessed. There was a random assignment of the participants to the conditions. Participants in the upward comparison condition were exposed to a scenario about a well-functioning teacher named Wil, which is a gender-neutral name in the Netherlands. This teacher was exceptionally well-functioning, accurate, conscientious, highly inspiring, always came up with exciting educational material, gave pupils a lot of personal attention, and kept order well, while his or her pupils found the atmosphere in his classes very pleasant and had very good results. In the downward comparison condition, the teacher was poorly functioning, not very accurate, conscientious or inspiring, did not come up with exciting material, did not give the pupils a lot of personal attention, and kept order poorly, while his or her pupils found the atmosphere in his classes not very pleasant and had inadequate results. 

In a pre-test among 30 teachers in secondary education, individuals randomly received either the upward or downward scenario. Participants rated their own functioning as worse than the upward target and as better than the downward target. Both means (*M*_upward_ = 2.00, *SD* = 0.54 and *M*_downward_ = 4.17, *SD* = 0.59) differed significantly from 3 (*t*(14) = −7.25 and 7.69, *p* < 0.01), which is the point at which one perceives oneself as equally good as the target. In the main study, the following measures were administered after the scenario, which could be answered on a 5-point scale ranging from *not at all* (1) to *very much* (5).

**Affect** [13]. Positive and negative affect were each measured with three questions. The items measuring positive affect were: “To what extent does this fragment give you a positive feeling?”; “To what extent do you find this fragment inspiring?”; and “To what extent do you find this fragment hopeful?” The negative-affect items were: “To what extent does this fragment give you a negative feeling?”; “To what extent do you find this fragment discouraging?”; and “To what extent do you find this fragment threatening?” Cronbach’s alphas for positive and negative affect were 0.85 and 0.74, respectively. Positive and negative affect correlated in the downward condition −0.43 (*p* < 0.01) and in the upward condition −0.45 (*p* < 0.01).

**Identification and contrast** [13]. Identification and contrast with the comparison target were measured with three items relating to identification and two items relating to contrast. The items measuring identification were as follows: “Do you recognize something of yourself in Wil?”; “Do you think you resemble Wil?”; and “Do you think that in the future you will reach (or will keep) Wil’s position?” The two items measuring contrast were: “Do you think that Wil is another kind of person than you?” and “Do you see a contrast between you and Wil?” In a principal components factor-analysis, one factor emerged with an Eigenvalue greater than 1 (3.62), explaining 72.5% of the variance. Therefore, it was decided to construct a single scale to measure identification versus contrast. The items measuring contrast were recoded, so that a high score on the identification-contrast measure indicated a high identification and a low contrast. The internal consistency of the scale was very high (Cronbach’s α was 0.90).

### 5.3. Data Analysis

To assess the influence of burnout and the moderating role of SCO on the affective responses to social comparison, positive and negative affect were regressed on the comparison direction (coded as −1 for the downward condition and +1 for the upward condition), burnout, SCO, and the interactions between these variables.

## 6. Results: Study 2

### 6.1. Affective Responses

Table 3 shows that upward comparison induced more positive and less negative affect than downward comparison. In addition, there was a significant association between burnout and negative affect, indicating that with increasing levels of burnout, individuals reported more negative affect after both upward and downward comparison. Increasing levels of burnout were associated with less positive affect, but this association was qualified by a significant interaction between burnout and comparison direction (see Figure 3). 

Individuals experienced less positive affect with increasing levels of burnout only after being confronted with an upward comparison target (upward, B = −0.29, *p* < 0.01 vs. downward, B = −0.01, *ns*). There was no interaction between the burnout and comparison direction on negative affect, showing that after both downward and upward comparison, individuals reported more negative affect the higher their level of burnout [13]. Hence, the expectation that individuals high in burnout would report less favorable responses (Hypothesis 1) was confirmed for both positive and negative affect following upward comparison, and for negative affect following downward comparison.

In line with Hypothesis 2, with increasing levels of SCO, individuals reported more positive affect following social comparison. Table 3 also shows an interaction effect between SCO and burnout on positive affect, but this effect was qualified by a three-way interaction. It was only in the upward comparison condition that the interaction between burnout and SCO was significant (for upward comparison, B = −0.19, *p* < 0.01, for downward comparison, B = 0.03, *ns*). As Figure 4 shows, only among individuals high in SCO, did upward comparison generate less positive affect as burnout was higher (high SCO, B = −0.47, *p* < 0.01, low SCO, B = −0.09, *ns*). Hence, the expectation that SCO would enhance the link between burnout and responses to social comparison (Hypothesis 3) was confirmed with regard to positive affect following upward comparison.

### 6.2. Identification–Contrast

To assess whether burnout influenced the identification–contrast processes, identification–contrast was regressed on the burnout and comparison direction. Table 3 shows that, overall, individuals identified themselves more (or contrasted themselves less) with the upward comparison target than with the downward comparison target. Moreover, in line with Hypothesis 1, we found a two-way interaction (see Figure 5), showing that individuals identified themselves less (or contrasted themselves more) with the upward target as burnout levels increased (B = −0.38, *p* < 0.01), whereas individuals identified themselves more (or contrasted themselves less) with the downward target as burnout levels increased (B = 0.28, *p* < 0.01) [13].

In the upward condition, burnout had significant associations with identification–contrast, as well as with positive affect (B = −0.29, *p* < 0.01) and negative affect in both conditions (B = 0.39, *p* < 0.01). Therefore, it was examined to what extent identification–contrast mediated the effects of burnout upon affect [13]. Therefore, regression analyses were performed in which identification–contrast and burnout were entered simultaneously (see [39]). These analyses were executed for both conditions separately. In the case that the effect of identification–contrast is significant and makes the effect of burnout non-significant, identification–contrast fully mediates the relationship. However, when it only reduces the effect of burnout, there is only partial mediation. As Table 4 shows, in the upward condition, identification–contrast had a significant main effect on positive affect, showing that positive affect was higher as there was more upward identification. The effect of burnout was no longer significant. This suggests that identification–contrast fully mediated the association between burnout and the experience of positive affect following upward comparison. The association between burnout and negative affect after upward and downward comparison was reduced slightly, and the effects of identification–contrast were significant, suggesting a partially mediating role of identification–contrast [13]. Hence, Hypothesis 4, which predicted that associations between burnout and the affective responses to social comparison would be mediated by identification and contrast, was fully confirmed for positive affect following upward comparison.

To explore whether SCO moderated the association between identification–contrast and affect, in each condition, positive and negative affect were regressed on identification–contrast and SCO and their interaction (see Table 4). Only the two-way interaction on positive affect following upward comparison was significant, indicating that identification–contrast with the upward target was more strongly related to positive affect among high-SCO individuals (B = 0.95, *p* < 0.01) compared with low-SCO individuals (B = 0.59, *p* < 0.01). The final model concerning the experience of positive affect from upward comparison is depicted in Figure 6 [13].

## 7. Discussion

### 7.1. Burnout

Two studies were conducted to determine the responses to upward and downward comparison among teachers with different levels of burnout [13]. It was expected that individuals in a state of burnout would respond less favorably to downward and upward comparison, that is, they would respond with more negative and less positive affect and would identify and contrast themselves in a less self-serving manner. It was expected that identification and contrast processes would be considered important in determining the affective responses to social comparison (e.g., [19]): identification with an upward target and contrast with a downward target may result in a favorable affective response, whereas downward identification and upward contrast would result in an unfavorable affective response. 

Study 1 retrospectively investigated how teachers responded to comparison with better-off and worse-off others, using measures for identification and contrast which included affect. The results of this first study were in line with the prediction that individuals interpret downward comparison information more negatively and interpret upward comparison information less positively with increasing burnout levels [13]. That is, with increasing levels of burnout, upward comparison evoked less identification and more contrast, and downward comparison evoked more identification.

A largely similar pattern emerged in Study 2, in which teachers were confronted with a scenario about a well-functioning or a poorly functioning colleague and in which identification versus contrast was assessed separately from affect. As burnout levels increased, individuals showed more negative affect after downward comparison, and less positive affect and more negative affect after upward comparison. In line with our expectation and with a similar study [40], as individuals were higher in burnout, they identified themselves less (i.e., contrasted themselves more) with the upward target and contrasted themselves less (i.e., identified themselves more) with the downward target. This may be explained by the fact that individuals who experience burnout are characterized by relatively low levels of perceived control. It has been suggested that unrealistically high expectations may trigger the development of burnout; thus, burned-out individuals may feel threatened in their sense of control [13,41].

The fact that the two rather different methods yielded largely similar conclusions about the responses to social comparison substantiates the validity of the two paradigms and the reliability of the results. In general, the present results are in line with those found in other studies (e.g., [28,42]), which indicated that downward identification and upward contrast were higher among those high in burnout. Moreover, both studies indicated that individuals high in burnout responded not only less positively, but also more negatively to upward comparison. These findings are also in line with other research on social comparison and burnout in organizations. Buunk, Van der Zee et al. [40], using the same method as in Study 2, showed that with increasing levels of burnout, nurses responded with more negative affect and less positive affect to a target who was performing better than themselves (see also [43]). Hence, it seems that individuals in a state of burnout tend to perceive upward comparison as a threat, maybe because it makes them aware of their inferiority. As has been suggested by numerous authors, being outperformed by a better-performing other may be threatening to one’s self-esteem and may therefore result in a negative mood and a low sense of well-being (e.g., [13,16,29]).

The finding from Study 2 that downward comparison generated less positive affect and more negative affect than upward comparison, independent of an individual’s level of burnout, seems not in line with Wills’ downward comparison theory [44], which emphasizes the beneficial effects of downward comparison for individuals experiencing a decline in well-being. It is possible that Wills’ downward comparison theory only applies under specific conditions. Buunk and Ybema [19] noted that individuals may benefit more from downward comparison with an abstract individual (e.g., in the form of a test score) than with an actual individual. Ranoff-Bulman reported that interviewing paralyzed individuals depressed her as it reminded her of her own vulnerability, realizing that misfortune could also happen to her [45]. Furthermore, there is evidence that downward comparison information is only helpful for individuals whose well-being is both permanently and temporarily threatened [46]. This suggests that burned out individuals may only benefit from downward comparison when their well-being is endangered quite strongly, such as when receiving a poor evaluation [13].

The results of Study 2 underline that positive and negative affect represent different psychological processes [13,36]. The results are in line with the notion that positive affect reflects the approach of rewarding situations, whereas negative affect reflects a withdrawal from threatening situations. As a positive response may be associated with opportunities and obtaining resources, upward comparison will mainly result in positive feelings, especially when individuals identify themselves with the comparison target. Downward comparison does not reflect something to aim at, but rather something to be avoided, and may therefore result primarily in negative affect, especially in the case of identification.

### 7.2. Identification and Contrast

Identification–contrast was expected to mediate the affective responses to social comparison among individuals high and low in burnout [13]. However, identification seemed to mediate only in part the effect of burnout on negative affect social comparison. Thus, although individuals high in burnout identified and contrasted themselves in a less productive way, this was not the only reason why they showed negative feelings after social comparison. In positive affective responses to upward comparison, identification fully mediated the association with burnout. Specifically, those high in burnout derived less positive affect from upward comparison because they identified themselves less with the other doing better. This finding is in line with the results from a study in which identification mediated the relationship between perceived control and positive affect after upward comparison [43]. The results are also in line with the notion that individuals with a low self-esteem may not expect to be similar to others doing well and may therefore not identify with these targets [20]. Thus, several studies suggest that individuals high in burnout or low in self-esteem have a relatively low capacity to identify themselves with others doing better.

An alternative interpretation for the finding that individuals high in burnout reported less upward identification can be that upward identification might serve as a buffer against burnout. That is, these individuals are in a state of burnout just because they are not able, for some reason, to identify with better-off others. Upward identification might foster feelings of superiority and might protect one against burnout. That is, individuals might derive a sense of self-efficacy from upward identification or might learn how to deal with difficulties and may in this way escape burnout (see [43]). However, the experiment in Study 2 was not designed to examine this reasoning, and the results are, as could be expected, more in line with the former interpretation. Further, prospective research would be necessary to examine the relationship between upward identification in general and the development of burnout symptoms [13].

### 7.3. Social Comparison Orientation

In both studies, it was expected and found that the affective responses to social comparison would be more pronounced among individuals with a high social comparison orientation (SCO), that is, among individuals with a strong dispositional need to compare themselves with others. It was also expected that SCO would enhance (moderate) the association between burnout and the response to social comparison information. It appeared that only the affective reactions to upward comparison were moderated by SCO. Only, or particularly, among individuals high in SCO, increasing levels of burnout resulted in less identification and more contrast in upward comparison (Study 1) and in less positive affect derived from upward comparison (Study 2). Although Study 1 suggested that both positive and negative responses to upward comparison depend on SCO, in Study 2, the experimental study showed that, after forced upward comparison, this is only true for the experience of positive affect. Both studies indicate, however, that SCO does not moderate reactions to downward comparison [13].

In Study 2, it was also explored whether SCO influenced the experience of positive affect after upward identification. Indeed, SCO appeared to enhance the amount of positive affect experienced from upward identification. This finding is theoretically quite important, because it suggests that not everyone may profit to the same extent from upward identification; in particular, those with a strong dispositional need for social comparison information may derive positive affect from upward identification. As these individuals seem to be characterized by feelings of uncertainty (see [13,14]), upward comparison may reassure them that they are indeed nearly as good as the upward target, whereas this information adds little to the sense of certainty of highly confident individuals. The present study thus adds to the growing literature documenting how social comparison may evoke quite distinct responses depending on SCO.

### 7.4. Limitations

The findings should be interpreted against some limitations. A limitation of our study concerns the design of our experimental study, which did not include pre-measurements of affect. Although these could be associated with levels of burnout, incorporating pre-measurements of affect would have allowed for more fine-grained analyses regarding the effects of our experimental manipulation. Furthermore, the use of self-report, cross-sectional data may lead to common method variance and may in this manner have influenced our results [45]. Although personal variables such as burnout, SCO, and affect–identification–contrast are difficult to assess with objective measures (see [47]) and although associations between self-report measures are not always upwardly biased [48], it would be important to replicate our findings in future studies.

It should also be noted that the amount of explained variance in the present studies, in particular the variance explained by the interaction effects, was not very high. However, one should realize that studies using ANOVA generally do not report R^2^, and the calculation of R^2^ often reveals values equal to or lower than those in the present research. More importantly, the use of explained variance as a measure of the importance of the findings may be problematic [49,50]. Rosenthal and Rubin [49] have shown that explained variance can be a deceptive measure by giving an example of a fictitious experiment (N = 100) in which a variable has important effects (30 vs. 70% survival following treatment) but accounts only for 16% of explained variance. In addition, several features of the present research may have resulted in the low explained variance [49]; for instance, the use of experimental designs, which generally explain less variance than correlational studies, and a modest number of predictors was used, which also reduces the amount of explained variance [49]. To conclude, a low amount of explained variance does not necessarily imply that the importance of the results is limited as well [13].

The present research has a number of potential practical implications. First, school principals should in general be careful when providing social comparison information to teachers, for example, on how well others are performing, as this might generate feelings of frustration and failure. However, for new teachers, this might be beneficial when the other is presented as a role model with whom new teachers may identify themselves, and they may learn from that other, more experienced colleague. Second, school principals should be sensitive to the fact that for teachers high in SCO, social comparison information may have different effects than those on employees low in SCO. It may sometimes be difficult to know whether someone is high or low in SCO, but this may become transparent when the individuals spontaneously mention comparisons with others or show a high level of insecurity. Often, it may be wise to discourage such people from making social comparisons, but instead to focus on how he or she is doing his or her work. Third, in general, it may be recommendable to present accurate information on how well-performing colleagues have attained their high level of performance. This might induce identification or make the high performance of others less threatening. To conclude, given the prevalence of social comparisons among teachers, this issue deserves due attention from school principals, managers, and human resources officers to prevent the potential destructive effects of social comparison on the well-being and motivation of teachers.

## 8. Conclusions

Individuals are often confronted with information about how others are doing, for instance, via the media, or through friends and family. Especially in the work situation, social comparison may be a quite prevalent phenomenon (e.g., [23,24,25]). As the present research indicates that comparison with others may evoke negative feelings, one could infer that frequent or continuous exposure to social comparison information may result in distress and may in the end induce feelings of burnout, particularly for those with a strong need to compare themselves with others. To be more specific, in the individual treatment of burnout, attention should be paid to dysfunctional social comparison processes. People who are negatively affected by social comparison information may be taught how to use social comparison information in a more adaptive way, for instance, by focusing on similarities with better-off others. As such, social comparison changes from a source of distress into a source of self-esteem enhancement [13].

## Figures and Tables

**Figure 1 ijerph-19-13139-f001:**
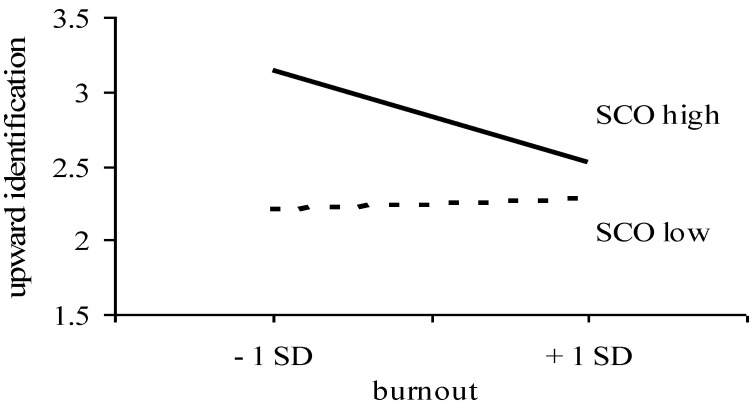
Upward identification as a function of burnout and SCO [13].

**Figure 2 ijerph-19-13139-f002:**
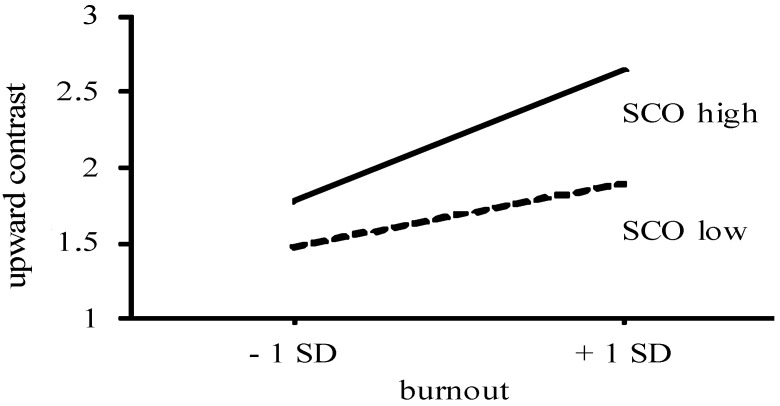
Upward contrast as a function of burnout and SCO [13].

**Figure 3 ijerph-19-13139-f003:**
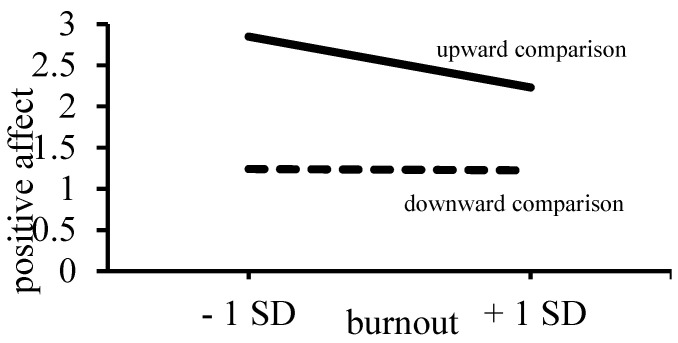
Positive affect as a function of burnout and comparison direction [13].

**Figure 4 ijerph-19-13139-f004:**
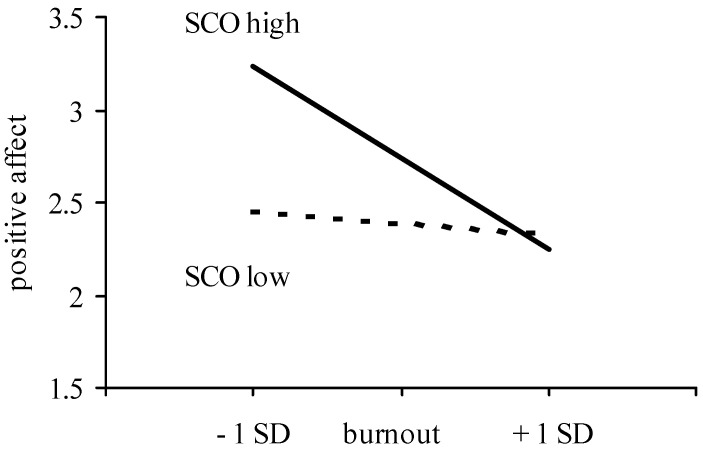
Positive affect after upward comparison as a function of burnout and SCO [13].

**Figure 5 ijerph-19-13139-f005:**
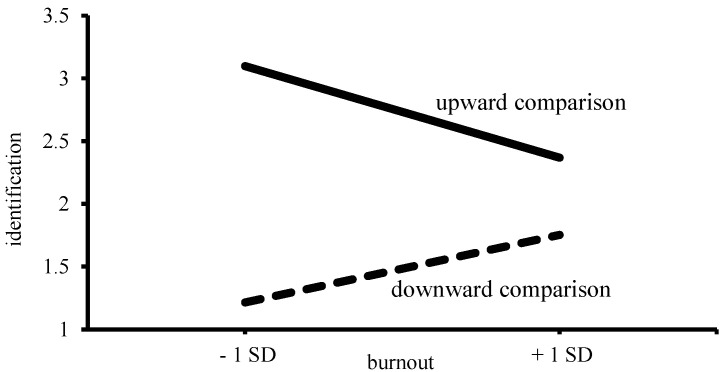
Identification as a function of burnout and comparison direction [13].

**Figure 6 ijerph-19-13139-f006:**
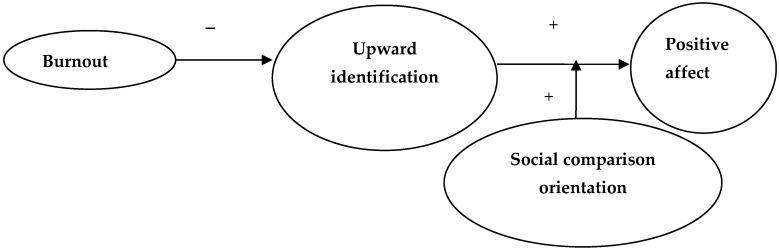
Predicting positive affect following upward social comparison [13].

**Table 1 ijerph-19-13139-t001:** Results of hierarchical regression of positive and negative affect as a response to upward comparison in Study 1 [13].

	Upward Comparison			
	Identification		Contrast	
	R^2^	B	R^2^	B
Step 1: Main effects	0.09 **		0.33 **	
Burnout		−0.17 *		0.39 **
SCO		0.28 **		0.35 **
Step 2: Interaction effect	0.04 *		0.02 *	
Burnout × SCO		−0.17 *		0.13 *

* *p* < 0.05; ** *p* < 0.01.

**Table 2 ijerph-19-13139-t002:** Results of hierarchical regression of positive and negative affect as a response to downward comparison in Study 1 [13].

	Downward Comparison			
	Contrast		Identification	
	R^2^	B	R^2^	B
Step 1: Main effects	0.12 **		0.16 **	
Burnout		0.02		0.24 **
SCO		0.34 **		0.26 **
Step 2: Interaction effect	0.01		0.01	
Burnout × SCO		−0.01		0.08

** *p* < 0.01.

**Table 3 ijerph-19-13139-t003:** Results of hierarchical regression of positive affect, negative affect and identification–contrast on burnout, SCO and comparison direction (Study 2) [13].

	Positive Affect		Negative Affect		Identification–Contrast	
	R^2^	B	R^2^	B	R^2^	B
Step 1: Main effects	0.43 **		0.40 **		0.44 **	
Burnout		−0.17 *		0.22 **		−0.08
SCO		0.12 *		0.05		0.10
Direction		0.64 **		−0.59 **		0.66 **
Step 2: Two-way interactions	0.04 **		0.00		0.11 **	
Burnout × SCO		−0.12 *		−0.06		−0.04
Burnout × direction		−0.15 **		0.06		−0.33 **
SCO × direction		0.06		0.04		0.06
Step 3: Three-way interaction	0.01 *		0.00		0.00	
Burnout × SCO × direction		−0.11 *		0.10		−0.03

* *p* < 0.05; ** *p* < 0.01.

**Table 4 ijerph-19-13139-t004:** Results of hierarchical regression of positive and negative affect after upward and downward comparison on identification–contrast and SCO [13].

	Upward Comparison				Downward Comparison			
	Positive Affect		Negative Affect		Positive Affect		Negative Affect	
	R^2^	B	R^2^	B	R^2^	B	R^2^	B
Step 1: Main effects	0.48 **		0.18 **		0.02		0.04	
Identification–contrast		0.73 **		−0.33 **		0.08		−0.31
SCO		0.05		0.14		0.05		0.08
Step 2: Interaction effect	0.03 *		0.02		0.00		0.00	
Identification–contrast × SCO		0.18 *		−0.11		0.03		−0.01

* *p* < 0.05; ** *p* < 0.01.

## Data Availability

The data are stored at Utrecht University data storage and may be accessed through the second author.

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
