# Peer review of "Burnout, Social Comparison Orientation and the Responses to Social Comparison among Teachers in The Netherlands"

_ijerph, 2022, doi:10.3390/ijerph192013139_

Round 1

Reviewer 1 Report

Comments and suggestions for authors to consider:

1.      Line 371: the sentence “It was“ appears uncompleted.

2.      The R square and beta coefficients in Tables 1, 2, 3, and 5 are in disorder, which makes the tables difficult to read.

3.      Table 4 cannot be found in the text.

4.      In section 2.2., the authors may create a table to more clearly present all the items of Identification-Contrast Scale mentioned in the paragraph.

5.      Were the two samples of the present study collected from two totally different teacher groups of secondary education? Did some of them participated in both studies? Was Study 2 designed and conducted based on the results of Study 1, or were they conducted synchronously? The authors may clarify the timeframe or relevance between them or how the conception of Study 2 was developed.

6.    It is commendable that the authors have spent quite a lot of time and efforts on conducting two studies with secondary education teachers. Rich discussions are provided for interpreting the research findings as well. However, readers may expect to learn more statements on academic, practical, or social implications recommended by the authors, which is supposed to be the most significant contribution this research brings about.

Author Response

  1. Line 371: the sentence “It was“ appears uncompleted.

Response: deleted, there was no sentence to be completed.

  1. The R square and beta coefficients in Tables 1, 2, 3, and 5 are in disorder, which makes the tables difficult to read.

Response: In my original manuscript as I submitted it, they were in order. Apparently something went wrong when entering it into the IJERPH system. I corrected it in the manuscript, and hope it stays in order after the resubmission. I couldn’t do this with track changes, as that didn’t show how the tables would look like, but now in any case the reviewer can judge the tables.

  1. Table 4 cannot be found in the text.

Response: Table 4 was mentioned in the text on page 15 (now page 12) There was also an incorrect reference to Table 5, which should be Table 4, and this has been corrected.

  1. In section 2.2., the authors may create a table to more clearly present all the items of Identification-Contrast Scale mentioned in the paragraph.

Response: The items were indeed not presented very clearly. Instead of putting them into a separate table, I have now made a sort of table in the text by much more clearly presenting the four scales, with all items listed on separate lines. I think this is as clear as presenting them in a table.

  1. Were the two samples of the present study collected from two totally different teacher groups of secondary education? Did some of them participated in both studies? Was Study 2 designed and conducted based on the results of Study 1, or were they conducted synchronously? The authors may clarify the timeframe or relevance between them or how the conception of Study 2 was developed.

Response:  As I have indicated now on page 7, these were indeed different samples, and no respondent participated in both studies. I also have described  the relation between both studies on page 7.

  1.   It is commendable that the authors have spent quite a lot of time and efforts on conducting two studies with secondary education teachers. Rich discussions are provided for interpreting the research findings as well. However, readers may expect to learn more statements on academic, practical, or social implications recommended by the authors, which is supposed to be the most significant contribution this research brings about.

Response: I have now, in the paragraph before the conclusion, added a rather extensive separate paragraph on practical implications.

Reviewer 2 Report

The submitted manuscript is a work in the field of psychology, which clearly confirms the existing data extensively described in the literature. In the reviewer's opinion, the design of the study is correct and the appropriate research tools have been selected. The methodical part is described correctly. However, the study group (both in Study 1 and Study 2) is very small for this type of research.

The work requires several corrections:

1. The abstract is written in an uninteresting way. I suggest you edit this part of the manuscript. 

2. The introduction section is too extensive and should be significantly shortened. 

3. In addition, there is no information about the methods of statistical analysis (these data should be placed before the result part). 

4. References should be corrected (e.g. 11, 12) in accordance with the requirements of IJERPH.

Author Response

The work requires several corrections:

  1. The abstractis written in an uninteresting way. I suggest you edit this part of the manuscript. 

Response: I have tried to make it more interesting

  1. The introductionsection is too extensive and should be significantly shortened. 

Response: I have reduced the Introduction considerably.

  1. In addition, there is no information about the methods of statistical analysis (these data should be placed before the result part). 

Response: I have done so, even though this is not at all common in psychological studies, and the nature of the statistical analysis was presented directly in the first sentence of the Results section. So, I moved this to the Method section

  1. References should be corrected (e.g. 11, 12) in accordance with the requirements of IJERPH

    Response: I have corrected reference 12, but I am sorry that I do not understand what is wrong with the other references.

Round 2

Reviewer 2 Report

After the changes made by the Autors, the work is much better.